# Mining-Related Metal Pollution and Ecological Risk Factors in South-Eastern Georgia

**Marika Avkopashvili** [1,2,*], **Guranda Avkopashvili** [3], **Irakli Avkopashvili** [4], **Lasha Asanidze** [5], **Lia Matchavariani** [1], **Alexander Gongadze** [3] and **Ramaz Gakhokidze** [1]

1. Faculty of Exact and Natural Sciences, Ivane Javakhishvili Tbilisi State University, Tbilisi 0179, Georgia; lia.matchavariani@tsu.ge (L.M.); rgakhokidze@gmail.com (R.G.)
2. Department of Geology and Environmental Science, University of Pittsburgh, Pittsburgh, PA 15260, USA
3. Elevter Andronikashvili Institute of Physics, Ivane Javakhishvili Tbilisi State University, Tbilisi 0177, Georgia; guranda.avkopashvili@tsu.ge (G.A.); alex.gongadze@gmail.com (A.G.)
4. Department of Earth Sciences, Ilia State University, Tbilisi 0162, Georgia; irakli.avkopashvili.1@iliauni.edu.ge
5. Vakhushti Bagrationi Institute of Geography, Ivane Javakhishvili Tbilisi State University, Tbilisi 0177, Georgia; lasha.asanidze@tsu.ge
* Correspondence: marika.avkopashvili@gmail.com or maa437@pitt.edu

**Abstract:** Gold and copper production is important to the Georgian economy, but at the same time, mineral resources are mined in one of the important agricultural areas of the country. This study evaluated water and soil quality in the region. Soil from 18 villages was analyzed. Some of these villages have not been investigated before and previous information about soil quality was unknown. Ecological risk factors and potential ecological risk were determined for the study area. Atomic absorption spectrometry was used to analyze heavy metals concentration in soil and water samples taken from the study area. Integrated water and soil data allowed us to see how these important natural resources influence each other. After the comparison of the four-year period of the study, we observed an increase of heavy metals increase in the soil in 2017 compared to 2014. Higher lead concentration was discovered within a two kilometer radius of the mining area whereas the highest cadmium concentration was observed in the village Ratevani, 15 km away from the nearest mine, where there was an extremely high ecological risk of Cd concentration. Further investigations are recommended to be done in Ratevani village as the people living in this village are at risk of Cd poisoning.

**Keywords:** soil pollution; mining; heavy metals; ecological risk; particulate matter

## 1. Introduction

Along with industrialization, mining activities also negatively influence the environment [1–3]. The increasing demand for mineral resources contributes to the growth of the mining industry in the world. Li et al. identified an association between industrialization and environmental quality [4]. Industrialization can have negative impacts on the environment [5]. Mining is one of the pathways by which various heavy metals (e.g., As, Cd, Cu, Mn, Pb, Zn) enter the environment [6]. Open-pit mining impacts natural ecosystems through both removal of soil and vegetation and burial beneath waste disposal sites [7]. Heavy metal contamination primarily comes from ore processing, tailings disposal, and wastewaters around mines [8].

Gold and copper open-pit mining in Kazreti (south-east Georgia) began in the 1970s and continues to play a significant role in the economic and social development of Georgia [9]. In 2011, a new pit called Abulbuqi was opened; in 2014 another pit, called Sakhdrisi quarry was opened and increased ore extraction from the pit and its further processing [10]. Geologically, the area is underlain by quaternary sediments, limestones, sandstones, and magmatic formations of the Cretaceous and Jurassic periods, and the main ore deposits

are barite-gold-copper-polymetallic volcanogenic massive sulfides. These deposits show consistent patterns of ore zonation. These zones are from the base upward: (1) copper-pyrite (chalcopyrite, pyrite, quartz, and minor sphalerite, chalcozine, epidote, anhydrite, galena, and fluorite), (2) barite-polymetallic ore (barite, sphalerite, pyrite, galena, and minor chalcopyrite and marcasite), and (3) gold-bearing quartzites (quartz, barite, calcite, and pyrite) [11,12].

At the processing plant, gold is extracted from the ore by the cyanide leaching method where piles of crushed ore are soaked with cyanide solution [13]. Ores are extracted using drilling and blasting methods, and therefore release large volumes of particles into the air. The particulate matter (PM) could transport both harmful compounds and heavy metals from other sites [14]. The PM10 μm can travel 50 km away through the wind [15], while PM2.5 μm can travel hundreds of kilometers away from the eruption point and be deposited onto the soil during precipitation, which pollutes soils in the Bolnisi and Dmanisi municipalities.

The study area is located in the Mashavera river basin, one of the important agricultural regions of the country where much of the food is supplied to Georgian markets. Mashavera river is the main irrigation source in many settlements of the region. The Kazretula river is one of the eastern tributaries of the Mashavera and flows next to the mines. Before the mining company built a water treatment system at the mining site in 2019, the industrial water used at the ore processing plant was being discharged into the Kazretula river and causing Cu, Cd, Mn, Zn, and Pb pollution that impacted the Mashavera river water quality downstream. During their research, Matchavariani and Kalandadze found excess amounts of Cu, Cd, Zn, and $SO_4^{2-}$ in the Kazretula river. They also found high quantities of Cu, Mn, and Zn in the soils situated on the banks of the Mashavera river [16]. Withanachchi et al. investigated water in the Mashavera river basin and found alarming levels of heavy metals contamination that exceeded national and international thresholds [17]. The food products produced in the Mashavera basin are an important source for Georgian markets. The agricultural fields are used to grow various cereals, vegetables, herbs, and fruits. Henningsen et al. analyzed various food crops grown in the fields of this region and concluded that Cd, Cu, and Zn uptake by plants, especially by spinach, was high and exceeded precaution values [8].

Conducting new research was needed in the region, as its residents intensively consume locally grown food products and are at risk of poisoning. After 2014, the mining area and ore production increased in the region. This study aimed to examine soil and water pollution levels in 18 villages of the region (in the Bolnisi and Dmanisi municipalities) to assess their (soil/water) quality and evaluate how the mining area expansion impacted on the soil and water pollution levels. Previous studies have observed Mashavera river contamination from mine wastewater discharges, demonstrating the crucial need for comprehensive environmental monitoring to protect human health. For pollution assessment, Ecological Risk Factor (ER) and Potential Ecological Risk Index (RI) frameworks were used according to Hakanson and others [18–26]. Soils were sampled in 2014 and 2017 to evaluate changes of Cd, Cu, Mn, Pb, and Zn concentrations. The research evaluates sources of soil pollution in the study area to determine processes creating these patterns in soil chemistry; similar studies can be found in other works [27–31]. Finally, interpolation was done to estimate pollution levels across the entire study area, related to the works of Hou et al. and Luzio et al. [32,33]. After comparing the four-year period of the study, we observed an increase in heavy metals in the soil in 2017 compared to 2014. It is remarkable that the highest Pb concentration was found in the soils near the mining area, and its concentration decreased going further from the mine, whereas the highest Cd concentration was found in the village Ratevani, which is located further away from the mines. This pattern has not been observed in previous studies in the region. In addition, some of the villages in the study area have not been investigated before. It is recommended to do further research in Ratevani village as its population are at risk of Cd poisoning.

## 2. Materials and Methods

### 2.1. Description of Study Area

The orographic scheme of study area (Figures 1 and 2) predominantly includes low elevations. The climate of Kazreti belongs to the steppe type of dry subtropics with moderately cold winters and hot summers [13]. The annual rainfall is on average 500–550 mm, and the average wind speed in 2014–2017 was 0.7 m/s, both north-east and south-west winds are widespread. The length of the Mashavera river is 66 km, the river run off varies over the seasons. The peak discharge is in spring—40% of annual runoff and decreasing through the winter as follows, 30.8% in the summer, 16.8% in the fall, and 12.4% in the winter [17]. The main soil types in the study area are chernozems, cinnamonic soils, and brown forest soils [10].

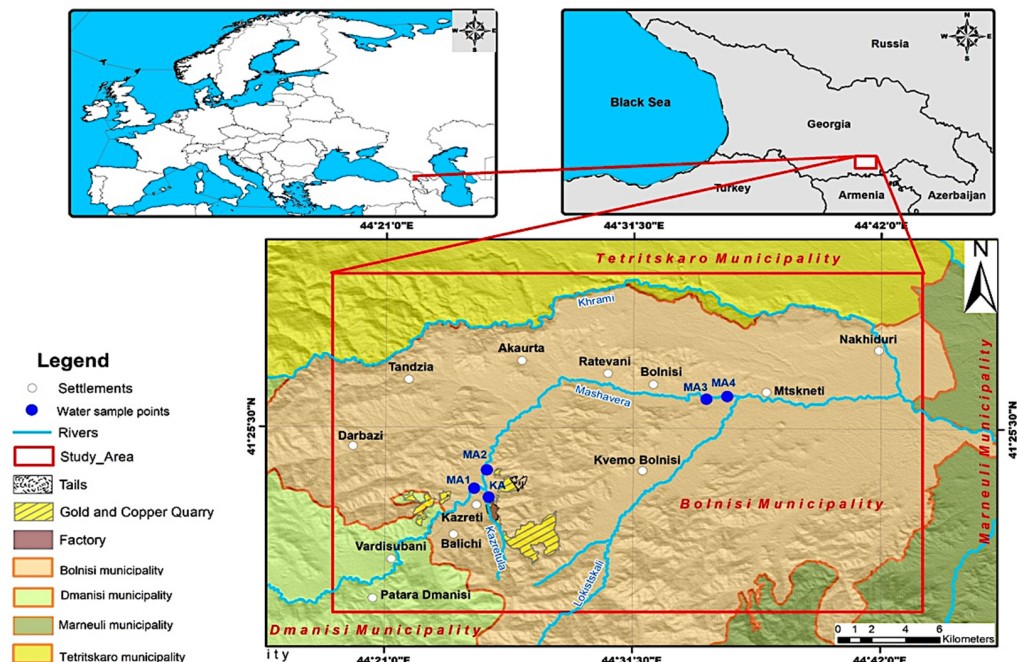

**Figure 1.** The map of the study area. Water sample points shown with blue dots (MA1 = Mashavera river before intersecting with Kazretula river; KA = Kazretula river; MA2 = Mashavera river at one km after intersecting with Kazretula river; MA3 = Mashavera river at 30 km after the intersect with Kazretula river; MA4 = at 35 km after intersect with Kazretula river).

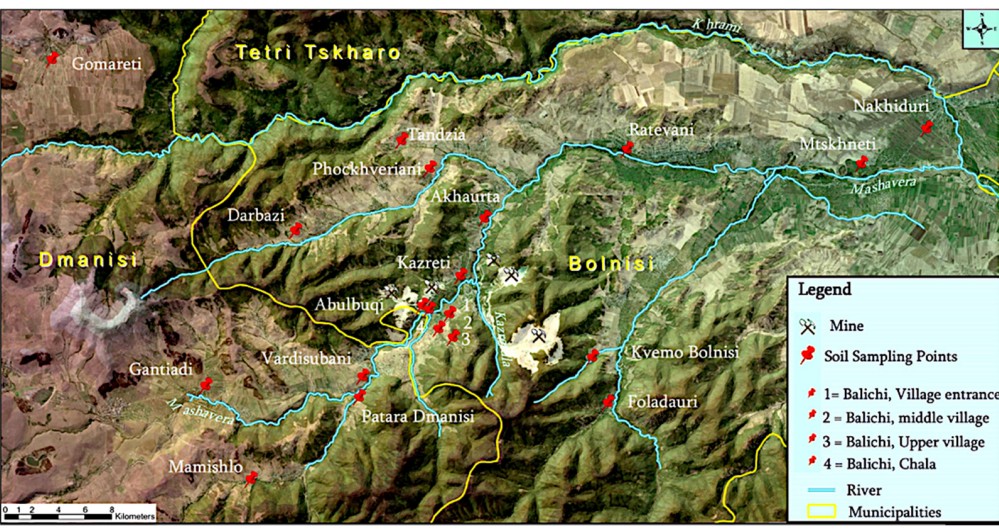

**Figure 2.** Soil sampling points in the study area.

## 2.2. Sampling and Analytical Methods

Soil Sampling: In 2014, soil samples were collected in the Bolnisi and Dmanisi municipalities from agricultural fields various distances away from the mining site (Figure 2). The sampling point closest to a mine was 0.5 km away, and the farthest was 30 km away. Repeat sampling was conducted in 2017, and soil samples were taken from the same points as in 2014. The sampling and analytical methodology were the same for both dates (2014–2017). Each sample represents a surface soil aliquot taken at 0–5 and 30–35 cm from the depth of the soil. Soil samples were taken with scoop samplers, which were washed between each sampling collection. The study area was divided into regular grids of 1000 × 1000 m, where the samples were collected at five points, mechanically mixed, and a composite sample was prepared. One composite sample was made for each sampling point. The composite soil sample was transferred into a polyethylene bag, labeled, and transported to the laboratory. Unwanted materials such as stones, leaves, and debris were removed from the soil samples. The samples were oven-dried at 105 °C for 24 h, followed by grinding and sieving using a 2 mm sieve to homogenize the sample. To determine the concentrations of metals, 5 mL of 65% $HNO_3$ (trace metal grade) was added to 1000 g of soil in a 50 mL volumetric flask. The flask was heated in a water bath (100 °C) for 2 h, then cooled down at room temperature for 15 min and filtered with a Whatman 0.45 μm paper filter into a 50 mL volumetric flask; the volume was made up to 50 mL with distilled water. The method used was similar to that of references [33,34]. These solutions were analyzed using AAS in the Iv. Javakhishvili Tbilisi State University Elevter Andronikashvili Institute of Physics, Georgia.

Water Sampling: Water samples were taken from Mashavera and Kazretula during July, October, and November. Water samples were collected in 1000 mL (HDPE) plastic bottles, which were washed and rinsed 3 times with distilled water prior to using them. Preservation of water samples was done by adding 2 drops of concentrated $HNO_3$ (trace metal grade) to each sample. Each bottle was labeled according to the sampling location and time. The samples were then put in an insulated box and taken to the laboratory for further processing. Water samples were filtered with Whatman 0.45 μm filter paper in a Buchner funnel and heated at 100 °C until the volume was reduced to 50 mL, in a normal volumetric flask 100 mL. The volume was made up to 100 mL with acidified water (3 mL concentrated $HNO_3$ + 1 L distilled water). Samples were analyzed by AAS in the chemistry laboratory of the Tbilisi State University Institute of Physics, Georgia.

Total concentrations were determined as soil and water concentrations added with their respective pseudo total digestions. For every 20 samples we included a replicate, a preparation blank, and an SRM. Montana soil SRM 2711a (National Institute of Standards and Technology, Gaithersburg, MD, USA) were used as mineral soil samples. All Cd, Co, Cu, Fe, Mn, Ni, Pb, and Zn concentrations for SRM materials were within 13% of their certified values. Preparation blanks had concentrations below detection limits for all metals.

## 2.3. Statistical Analyses

Ecological Risk Factor (ER) and Potential Ecological Risk Index (RI) were calculated for the assessment of soil contamination [33]. ER = $T_r^i$ × Cf, where $T_r^i$ is the toxic-response factor for a given substance (see Table 1) developed by Håkanson [18], and Cf is the contamination factor. The ratio between the total metal content in the soil sample (Cm) and world average shale values, as a background values, were used for the Cf calculation: Cf = $C_{metal}/C_{shale}$. See the grading standards for the ER and RI in the Table 2 [28,29].

The Pearson correlation coefficient was used to determine the correlation between the two variables: 1. The concentration of trace metals in soil and 2. the distance of each sampling point from the pollution source.

ArcGIS 10.2.1 was used to create hydrological modeling and Inverse Distance Weighted (IDW) interpolation maps of the study area. The hydrological tool allows us to determine flow direction, and show the stream network derived from the elevation model from SRTM data sources. Interpolation allowed to estimate values at new locations based on measurements found in a collection of points [32,33].

**Table 1.** Shale values and toxic-response factor.

|  | Shale Values mg/kg | Toxic-Response Factor for Metals |
|---|---|---|
| Pb | 20 | 5 |
| Mn | 850 | 1 |
| Cd | 0.3 | 30 |
| Zn | 95 | 1 |
| Cu | 45 | 5 |

**Table 2.** ER and RI formula descriptions and pollution levels.

|  |  | Description | Pollution Level |
|---|---|---|---|
| Ecological risk factor (ER) | $ER = T_r^i \times C_f$ | $C_f$ = Contamination factor $T_r^i$ = Toxic-response factor for metals | ER < 40 = low ecological risk; 40 < ER < 80 = moderate risk; 80 < ER < 160 = considerable risk; 160 < ER < 320 = high risk; ER > 320 = extremely high risk |
| Potential ecological risk index (RI) | $RI = \sum_{i=1}^{n} ER$ | ER–Ecological risk factor; n–Sum of the elements | RI < 150-minimal risk; $150 \leq RI < 300$-moderate risk; $300 \leq RI < 600$-considerable risk; $RI \geq 600$-extremely high risk |

## 3. Results and Discussions

### 3.1. Soil Chemistry

Cd, Cu, Pb, Mn, and Zn contents were determined in the soil samples taken in 2014 from several settlements of Bolnisi and Dmanisi municipalities (Table 3). In 2014, Cd, Cu, and Zn concentrations in the soils of the villages Nakhiduri and Mtskhneti (30 km from the nearest mine) were higher than in the soils of the villages near the mining area [34,35]. In 2014, Cu concentration in Nakhiduri village soils exceeded the permissible limit by 229 mg/kg, and zinc by 48 mg/kg (Table 3). The Mn content in almost all samples was above the permissible limit, and the highest Mn (660 mg/kg) was found in Nakhiduri. Although, the distance from Nakhiduri to the closest mine is much greater (30 km) than other villages, Cd, Cu, Mn, and Zn concentrations in the soil were higher than the rest of the villages and exceeded the allowable limits. In the other villages, metal concentrations in soils were within the permissible limits (a slight excess of Mn was observed in some places). During the open pit mining, the drilling and blasting methods used for ore extraction lead to the eruption of harmful compounds into the air (Figure 3). These chemicals travel long distances and cause air pollution in the Bolnisi and Dmanisi municipalities, they are deposited into the soil and cause its pollution [36–39].

In 2014, the agricultural crops in Nakhiduri and Mtskhneti village were intensively irrigated by the Mashavera river which was being polluted by Cu, Cd, and Zn [8,15]. Using contaminated water for the irrigation of the fields resulted in the pollution of soils by Cd, Cu, and Zn. In the other investigated villages, soil samples were taken from the agricultural fields where the Mashavera river is not used for irrigation. Therefore, the higher concentrations of heavy metals in Nakhiduri and Mtskhneti villages in 2014 is explained by the use of contaminated Mashavera river water for the irrigation of agricultural crops.

In 2017, Cd, Cu, Mn, Pb, and Zn were analyzed in 18 villages (Table 3). Compared to 2014, their average concentration in soil increased. Cd in soils increased by 2 mg/kg, Pb increased by 35 mg/kg, Zn increased by 20 mg/kg, and Mn increased by 9 mg/kg. In contrast, the average content of Cu, in 2017, decreased by 29 mg/kg, due to the noticeable decrease of Cu in Nakhiduri soils. In 2017, the Cu concentration in soils of the villages Nakhiduri, Mtskhneti, Foladauri, kvemo Bolnisi, Akhaurta, and Patara Dmanisi decreased by 208, 94, 23, 8, 15, and 3 mg/kg, respectively, compared to 2014. Zn concentration decreased by 165, 111, 12, 70, 40, 38, 11.6, and 5 mg/kg in the villages Nakhiduri, Mtskhneti,

Foladauri, Akhaurta, Gantiadi, Vardisubani, Mamishlo, and Patara Dmanisi, respectively. In contrast, a noticeable increase of Cu and Zn concentrations was observed in the villages located near the mining area (Balichi, Kazreti). Cd, Mn, and Pb concentrations in soils in Nakhiduri, Mn and Pb in Ratevani, Mn in Akhaurta, Mamishlo, and Patara Dmanisi were higher in 2014. Their Cd, Mn, and Pb concentrations increased in other investigated villages, the highest levels were discovered in the areas closest to the mine.

**Table 3.** In 2014–2017, elements concentration in soil and maximum, minimum, average, and standard deviation (mg/kg). [1] Permissible limits are taken from the Order of the Minister of Labor, Health and Social Affairs of Georgia, 2001. [2] Distance from the nearest mine to the sample points in kilometers. [3] Maximum allowable limits of heavy metals in soils according to the World Health Organization [40].

| Sampling Points | Cd | | Mn | | Cu | | Zn | | Pb | | |
|---|---|---|---|---|---|---|---|---|---|---|---|
| | mg/kg | | mg/kg | | mg/kg | | mg/kg | | mg/kg | | Dist. to |
| | 2017 | 2014 | 2017 | 2014 | 2017 | 2014 | 2017 | 2014 | 2017 | 2014 | Mine, km [2] |
| Nakhiduri | 2 | 2.34 | 746.4 | 660 | 153.3 | 361 | 182.4 | 348 | 8.9 | 19.6 | 30 |
| Mtskhneti | 2.6 | 1.44 | 504.3 | 538 | 91.2 | 186 | 122.6 | 233 | 20.4 | 17.2 | 22 |
| Foladauri | 2.2 | 0.14 | 527.2 | 487 | 30 | 53 | 59.7 | 72 | 13.7 | 8.6 | 12 |
| Kvemo Bolnisi | 1.5 | 0.17 | 571.4 | 503 | 24.1 | 32 | 58.8 | 55 | 15.2 | 15.8 | 11 |
| Ratevani | 4.8 | 0.63 | 516.1 | 559 | 145.7 | 149 | 619.7 | 155 | 14 | 20 | 14 |
| Mushevani | 1.5 | 0.24 | 629.8 | 345 | 42.9 | 32 | 72.7 | 56 | 25.5 | 19.7 | 9 |
| Akhaurta | 3.2 | 0.34 | 334.9 | 576 | 25.5 | 41 | 43 | 113 | 28 | 21.7 | 7 |
| Gomareti | 2.9 | 0.11 | 481.4 | 423 | 27.3 | 21 | 40.5 | 34 | 25 | 11 | 22 |
| Gantiadi | 1.9 | 0.16 | 582.2 | 560 | 27.4 | 28 | 44.3 | 84 | 18.4 | 10.4 | 10.5 |
| Vardisubani | 1.7 | 0.17 | 617.5 | 575 | 32.5 | 27 | 58.1 | 96 | 34.5 | 18 | 3 |
| Mamishlo | 1.2 | 0.12 | 466.7 | 546 | 26.1 | 24 | 54.4 | 66 | 35.1 | 9.7 | 10.5 |
| Patara Dmanisi | 1.2 | 0.14 | 494.3 | 642 | 25.8 | 29 | 66.1 | 71 | 39.5 | 10.1 | 5.5 |
| Abulbuqi | 2.2 | 0.28 | 712.2 | 567 | 168.3 | 88 | 628.3 | 120 | 80.7 | 16.6 | 0.1 |
| Tandzia | 2.5 | - | 605.5 | - | 85.2 | - | 59.6 | - | 57.8 | - | 8.5 |
| Darbazi | 2.2 | 0.15 | 550.2 | 420 | 32.3 | 28 | 70.1 | 58 | 62.8 | 13.4 | 6.5 |
| Photskhveriani | 2.2 | - | 616.5 | - | 45 | - | 72.1 | - | 68.5 | - | 7 |
| Kazreti | 2.8 | 0.17 | 600.2 | 462 | 66.3 | 32 | 82.5 | 66 | 91.3 | 16.2 | 3.5 |
| Balichi 1 | 2.9 | 0.18 | 572.1 | 525 | 145.5 | 87 | 93.4 | 78 | 86.9 | 28 | 2 |
| Balichi 2 | 3.1 | 0.20 | 499.1 | 509 | 109.2 | 98 | 233 | 87 | 73.3 | 21 | 2.5 |
| Balichi 3 | 1.3 | 0.22 | 609.3 | 515 | 65.4 | 101 | 120.5 | 94 | 98 | 18.7 | 3 |
| Balichi 4 | 3.3 | 0.3 | 671.5 | 523 | 142.6 | 124 | 300.6 | 89 | 90.1 | 18 | 1 |
| Average | 2.5 | 0.6 | 567.3 | 558.3 | 75.6 | 105 | 154.4 | 134 | 47.7 | 16.6 | |
| Standard Deviation | 1 | 0.73 | 93.4 | 59.9 | 53 | 106 | 171.6 | 91.8 | 30 | 3.8 | |
| Maximum | 4.7 | 2.3 | 746.4 | 660 | 168.3 | 361 | 628.2 | 348 | 98 | 21.7 | |
| Minimum | 1.2 | 0.1 | 334.9 | 462 | 24.1 | 27.5 | 40.5 | 54.8 | 8.9 | 10.1 | |
| Permissable Limits | Cd | | Mn | | Cu | | Zn | | Pb | | |
| Georgian Guidelines [1] | 2 | | 500 | | 132 | | 300 | | 32 | | |
| WHO [3] | 0.003 | | 2 | | 0.1 | | 0.3 | | 0.1 | | |

In the villages closest to the mine, the increase of Cd, Cu, Mn, Pb, and Zn in soils is a result of the expansion of the mining area and increase of ore production which contributed to releasing a higher amount of dust particles into the air. In the villages where Cu, Mn, and Zn reduction was observed, the agricultural fields where soil samples were taken were not used for cropping; the farmers fallowed soil between 2014 and 2017; the contaminated river water was not used to irrigate the soil and the metals could be leached to the lower levels of soils.

In the village of Nakhiduri in 2014, the Ecological Risk Factor (ER) for Cd was at 234 which is a high risk, and in Mtskhneti it was 144, which is a considerable risk (Table 4). In the other villages, the ER level was low. Accordingly, RI in Nakhiduri was 289 and in Mtskhneti it was 182.6, both at moderate risk. In other villages, there was a minimal RI.

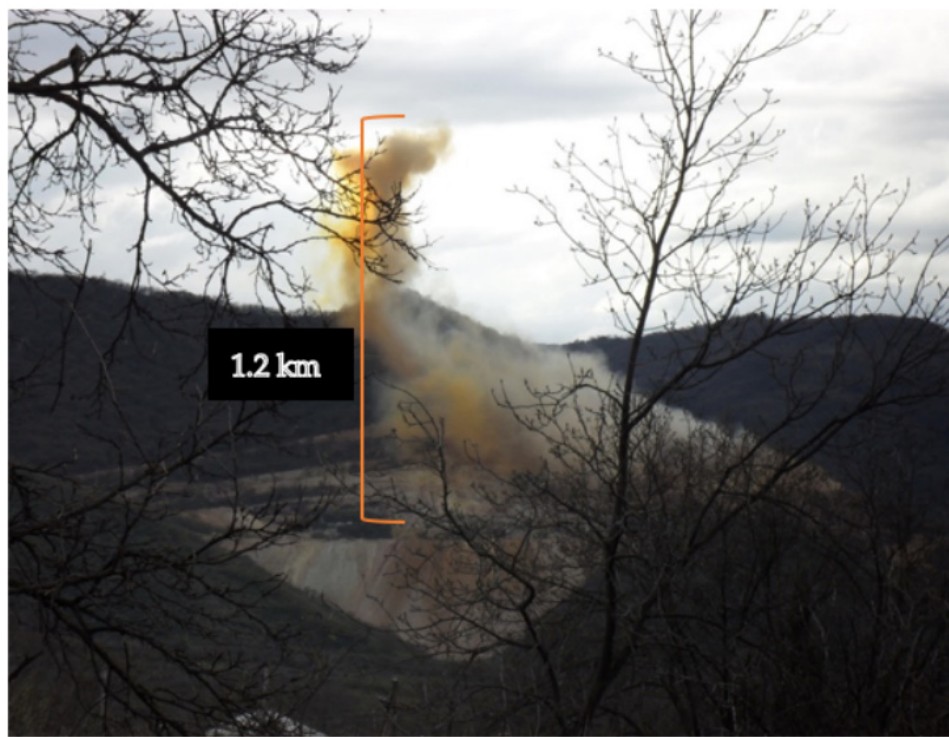

**Figure 3.** Explosion at the quarry and scattering dust particles through the wind into the environment, in 2011. Erupted dust particles are deposited onto the soil and contribute to its pollution. The scale of the dust eruption is about 1.2 km.

**Table 4.** Ecological Risk Factor (ER) and Potential Ecological Risk Index (RI) in 2014 and in 2017.

| Sampling Points | Ecological Risk Factor | | | | | | | | | | | |
|---|---|---|---|---|---|---|---|---|---|---|---|---|
| | Cd | | Cu | | Mn | | Zn | | Pb | | Potential Ecological Risk Index (RI) | |
| | 2017 | 2014 | 2017 | 2014 | 2017 | 2014 | 2017 | 2014 | 2017 | 2014 | 2017 | 2014 |
| Nakhiduri | 202 | 234 | 17 | 40 | 0.9 | 0.8 | 1.9 | 3.7 | 2.2 | 4.9 | 224 | 289 |
| Mtskhneti | 264 | 144 | 10 | 21 | 0.6 | 0.7 | 1.3 | 2.5 | 5.1 | 4.3 | 281 | 183 |
| Ratevani | 476 | 63 | 16 | 17 | 0.6 | 0.7 | 6.5 | 1.6 | 3.5 | 5.0 | 503 | 94 |
| Akaurta | 321 | 34 | 3 | 5 | 0.4 | 0.7 | 0.5 | 1.2 | 7.0 | 5.4 | 331 | 54 |
| Gantiadi | 193 | 16 | 3 | 3 | 0.7 | 0.7 | 0.5 | 0.9 | 4.6 | 2.6 | 202 | 26 |
| Dmanisi | 120 | 14 | 3 | 3 | 0.6 | 0.8 | 0.7 | 0.7 | 9.9 | 2.5 | 134 | 25 |
| Abulbuqi | 224 | 28 | 19 | 10 | 0.8 | 0.7 | 6.6 | 1.3 | 20.2 | 4.1 | 271 | 50 |
| Tandzia | 253 | - | 10 | - | 0.7 | - | 0.6 | - | 14.5 | - | 278 | - |
| Kazreti | 275 | 17 | 7 | 4 | 0.7 | 0.5 | 0.9 | 0.7 | 22.8 | 4.1 | 307 | 33 |
| Balichi | 326 | 22 | 16 | 11 | 0.8 | 0.6 | 3.2 | 1.0 | 22.5 | 4.7 | 368 | 46 |
| Kvemo Bolnisi | - | 17 | | 4 | | 0.6 | | 0.6 | | 4.0 | | 32 |

In 2017, ER in the observed villages increased dramatically for Cd (Table 4). In most villages, there was a high ecological risk level for Cd pollution; the maximum was observed in the village Ratevani, where the ER for Cd was 475.8, which means an extremely high risk according to the ER formula grading standards. Observing the peak Cd concentration in the village Ratevani is unclear and needs further investigations. The study area had a low ER for Cu, Mn, Pb, and Zn pollution. Moderate RI risk was shown in the examined villages. Despite there being no ecological risk for Pb pollution, its concentration increased fivefold in 2017 in the village Balichi. Such an increase of Pb concentration in soils within four years is alarming. Comparing the years 2014 and2017, Landsat and Google Earth images show the growth of the mining area (Figure 4). In 2014, the Sakhdrisi mine was opened, increasing ore extraction; intensive deposit blasting contributed to releasing more particulate matter into the air daily and resulted in the increase of Cd and Pb concentrations in the study area

in 2017 year. In 2010, soil pollution in the Bolnisi municipality was studied by Kalandadze and Matchavariani, who confirmed its contamination by Cd, Cu, and Zn (2019) [35].

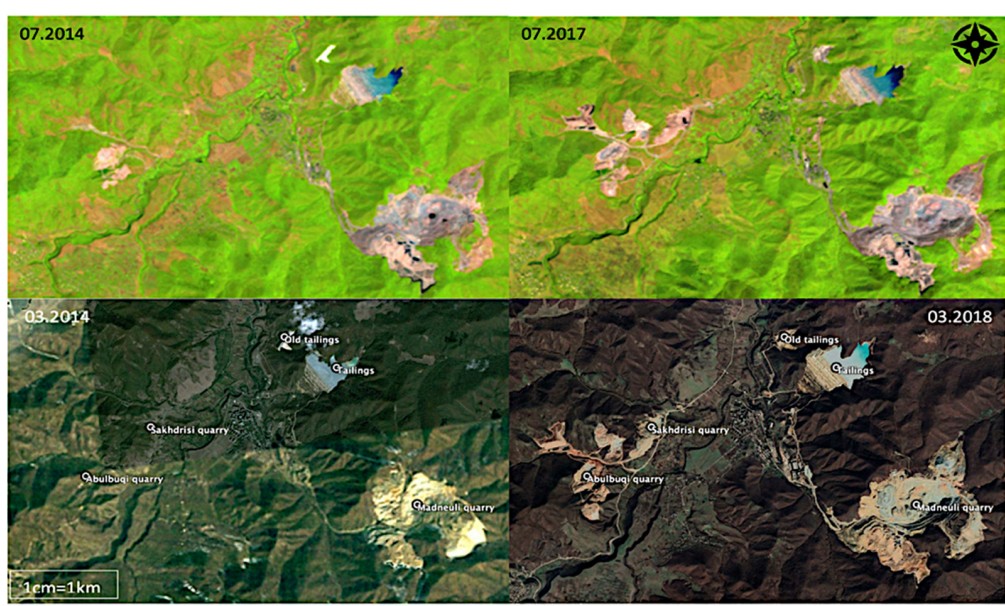

**Figure 4.** Landsat (**above**) and Google Earth (**below**) images, comparison of 2014 and 2017 mining area.

*3.2. Water Chemistry*

The Mashavera river is used to irrigate agricultural fields in the study area. The river is polluted by wastewater from mining discharged to the Kazretula river, which contributes soil pollution (Figure 5) [40–44]. The study area is hilly, the average elevation of the open-pits are 800 m, whereas the elevation of Mashavera river and agricultural fields around it are 100 m less than mines. Moreover, waste materials from the ore production are deposited near the mining area, which creates steep slopes, which are not terraced or planted to stabilize the slopes. The degree of mine hill slopes ranges from 25 to 57 degrees, whereas the slope of the agricultural fields ranges from 0 to 6 degrees (see the slope map on Figure 6). Therefore, runoff from the mine hill slopes leads to erosion and the contamination of nearby rivers and fields by heavy metals containing fines. The drainage network was derived from elevation data using the ArcGIS hydrological model (Figure 7). First-order streams are dominated by overland flow of water, they have no upstream concentrated flow, so they are most susceptible to non-point source pollution problems.

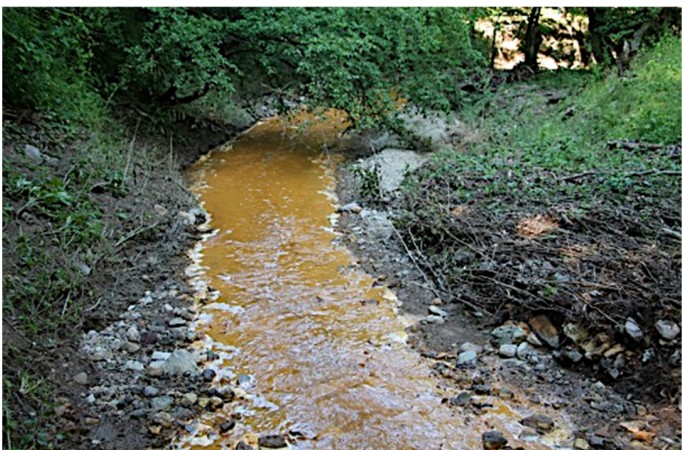

**Figure 5.** River Kazretula, one of the tributaries of the river Mashavera that is used for soil irrigation. Color change of the river indicates its pollution.

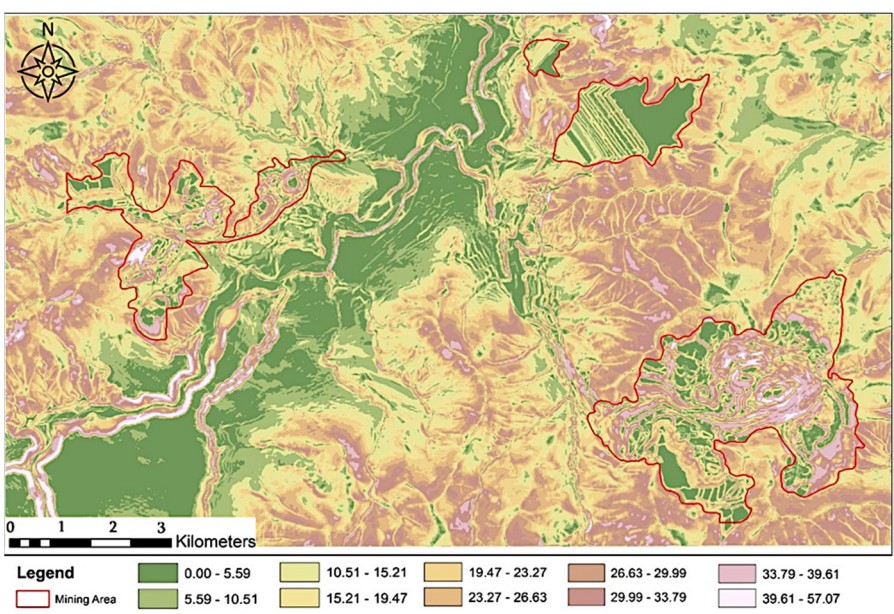

**Figure 6.** The slope map of the mining area in degrees.

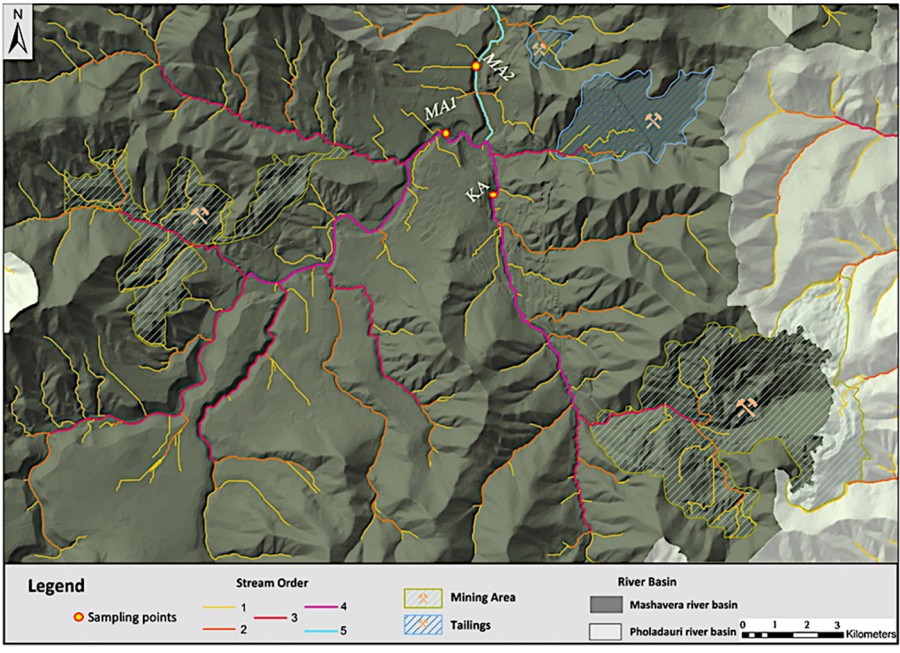

**Figure 7.** Streams flow direction around the mining area (MA1 = Mashavera river before intersecting with Kazretula river; KA = Kazretula river; MA2 = Mashavera river at one km after intersecting with Kazretula river.

We took five samples from the Kazretula (KA) and Mashavera (MA) rivers in July, October, and November in 2017 and analyzed Cd, Co, Cu, Fe, Mn, Ni, Pb, and Zn using an AAS (Atomic Absorption Spectrometer). Cd, Cu, Fe, Pb, and Zn were higher in October than in July, unlike Co, Mn, and Ni (Figure 8, Table 5); The highest amount of Cd was observed in November. Changing runoff of the rivers during the seasons also influenced the change of the metals' concentrations in the samples. The discharge of the Mashavera river was nearly halved in autumn (16.8% of annual discharge) compared to summer (30.8%). The decrease in the runoff of river water led us to observe higher concentration of metals in October and November, compared to July. The concentration of Cu, Fe, Mn, and Zn were higher in the Kazretula river, while Cd in October to July and Pb in October were

higher in the Mashavera river at 1 km after intersecting with Kazretula river. However, if industrial water is discharged into the Kazretula river, why is the concentration of Cd and Pb higher in the Mashavera river after the intersection with Kazretula river? This fact can be explained by the Tailings Pond, which is before the MA2 sampling point and is the source of acid mine drainage with pH values < 3 and high dissolved heavy metals concentrations (Table 6). In 2017, two acid mine drainages were analyzed and measured Cd, Cu, and Zn concentrations that showed high toxic values. These acid mine drainages flowed into the river and caused its contamination with toxic substances.

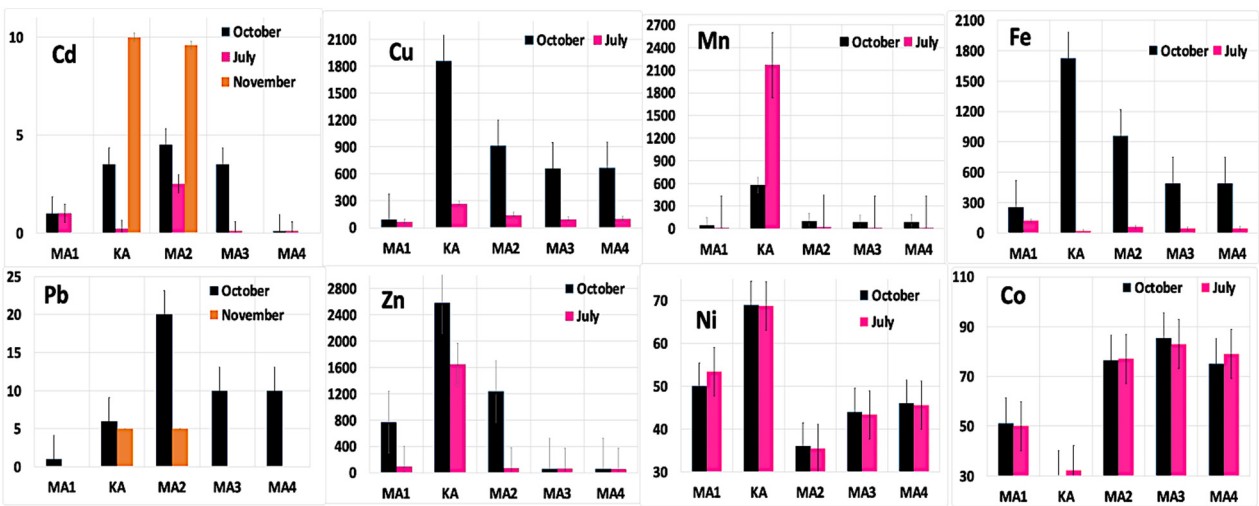

**Figure 8.** Cd, Cu, Mn, Fe, Pb, Zn, Ni, and Co content in the rivers (μg/L). Sampling points MA1 = Mashavera river before intersecting with Kazretula river; KA = Kazretula river; MA2 = Mashavera river at 300 m after intersect with Kazretula river; MA3 = Mashavera river at 5 km after the intersect with Kazretula river; MA4 = at 10 km after intersect with Kazretula river.

**Table 5.** In 2017, heavy metals minimum, maximum, median, and standard deviation in Mashavera and Kazretula rivers (μg/L). [1] '-' means not measured. [2] Permissible limits are according to Georgian official guideline values for water standards. [3] Maximum allowable limits of heavy metals in irrigation water according to the World Health Organization [40].

|  |  | Cu μg/L | Pb μg/L | Zn μg/L | Mn μg/L | Fe μg/L | Ni μg/L | Cd μg/L | Co μg/L |
|---|---|---|---|---|---|---|---|---|---|
| Permissible limits [2] |  | 1000 | 30 | 1000 | 100 | 300 | 130 | 1 | 130 |
| WHO [3] |  | 0.017 | 0.065 | 0.20 | 0.20 | 0.50 | 1.40 | 0.01 | 0.05 |
| Min | Oct | 92.1 | 1.0 | 57.3 | 41.2 | 254.4 | 36.0 | 0.1 | 32.3 |
|  | July | 58.0 | - [1] | 54.0 | 1.0 | 17.0 | 35.4 | 0.1 | 32.3 |
| Max | Oct | 1855.3 | 20.0 | 2581.9 | 576.1 | 1723.6 | 69.0 | 4.5 | 85.5 |
|  | July | 263.0 | - | 1648.0 | 2167.0 | 120.0 | 69.0 | 2.5 | 85.5 |
| Median | Oct | 667.9 | 10.0 | 768.9 | 84.3 | 487.6 | 46.0 | 3.5 | 76.5 |
|  | July | 96.0 | - | 65.0 | 1.0 | 45.0 | 46.0 | 0.2 | 77.0 |
| St.D. | Oct | 643.9 | 7.0 | 1045.2 | 224.6 | 585.4 | 12.3 | 1.9 | 22.4 |
|  | July | 80.2 | - | 706.7 | 967.2 | 38.6 | 12.7 | 1.0 | 22.5 |

**Table 6.** pH and heavy metals concentration in acid mine drainage of tailings pond. Samples were taken in 2017.

| Acid Mine Drainages | pH | Cu mg/L | Zn mg/L | Cd mg/L |
|---|---|---|---|---|
| Drainage 1 | 2.62 | 537.3 | 5010.0 | 23.65 |
| Drainage 2 | 2.97 | 165.3 | 4460.8 | 2.0 |

The Mashavera river is used to irrigate soils in the of villages Nakhiduri and Mtskneti, so it can pollute agricultural soils that are transferred into the food chain and impact human health. The higher concentration of heavy metals in Nakhiduri and Mtskhneti in 2014 was caused by use of the contaminated river Mashavera for soil irrigation. Due to the favorable climate conditions (warmer seasons) in the villages of Nakhidui and Mtskhneti, farmers cultivate crops twice a year, thus they intensively irrigate fields and use pesticides, which is one of the contributing factors impacting higher concentration of heavy metals in these villages. In contrast, in 2017, we observed heavy metals reduction in the villages of Nakhiduri and Mtskhneti, which was explained by the fact that the farmers were fallowing soils in 2014–2017, which prevented addition of heavy metals into the soils through the contaminated river water, and soil heavy metals were leached to the lower levels of the soil.

The water results showing high Cd, Cu, Fe, Mn, and Zn concentrations in the Kazretula river that indicates the influence of industrial activities on it. Seasonality also impacts the fluctuations of metals concentrations in the water. Maximum concentrations of Cd, Cu, Mn, Fe, and Zn were observed in the Kazretula river that also exceeded the permissible limits (Table 5). Intensive monitoring of water is recommended to be done in the study area as these rivers are valuable for the region's population.

The use of the correlation coefficient in the present study allowed us to determine the relationship between the heavy metal contamination in soils and the distances of the sampling points from the source of contamination. The correlation coefficient was calculated separately for each investigated element. The two different variables were (1) the content of Cd, Cu, Mn, Pb, and Zn in the soils, and (2) the distance from each sampling point to the source of contamination, in this case to the open pit.

The correlation coefficient was calculated for Cd, Cu, Mn, Pb, and Zn, analyzed in 2017, to determine whether the distance from the mine to the village affects the change in the concentration of heavy metals in the soils. Out of the five elements, we obtained a high correlation only for Pb, r (Pb) = 0.73, which means that the closer the village is to the mining area, the higher the lead content is in the soils of this village, and vice versa, the farther away the village is from the quarry, the lower the Pb content in the soils is. No similar pattern was found for the other elements, r (Cd) = −0.06, r (Cu) = 0.02, r (Mn) = 0.05, r (Zn) = 0.12, (Table 7). In the case of Cd, Cu, Mn, and Zn, there was no correlation. For example, Cd content in the soil of the village Gomareti was almost the same as in the village Balichi, even though the village Gomareti is located 22 km away from the mine, and the village Balichi is 2 km away. A positively high correlation was found only for Pb.

**Table 7.** Correlation coefficient between the heavy metals' contamination in soils and the distances of the sampling points from the source of contamination, in 2017. (Correlation ranges +1 high correlation to −1 no correlation).

|  | **Cu** | **Mn** | **Zn** | **Pb** | **Cd** |
|---|---|---|---|---|---|
| Correlation | 0.02 | 0.05 | 0.12 | 0.73 | −0.06 |

The interpolation maps were created by using ArcGIS (IDW), they show the distribution of Pb (Figure 9) and Cd (Figure 10) in soils in the Bolnisi and Dmanisi municipalities. In Figure 9, the colors range from blue to red, indicating the minimum and maximum Pb contents in the soils, i.e., the dark red color indicates the highest level of Pb contamination, while the dark blue shows no Pb contamination in the soils. In Figure 10, the colors range from green to red, which is similarly the minimum and maximum Cd content in soils. As can be seen from the maps, the highest level of Pb pollution presented in the villages of Balichi and Kazreti, which are the closest villages to the mine. The highest Cd concentration was found in the village Ratevani, which is 15 km away from the mining area. Cd content was less in the villages adjacent to the mining area. It is unclear why the Cd level was higher further away from the mine rather than closer to it. Further investigation of Ratevani

soils is highly recommended because high cadmium concentration is a risk for the human health of the village population.

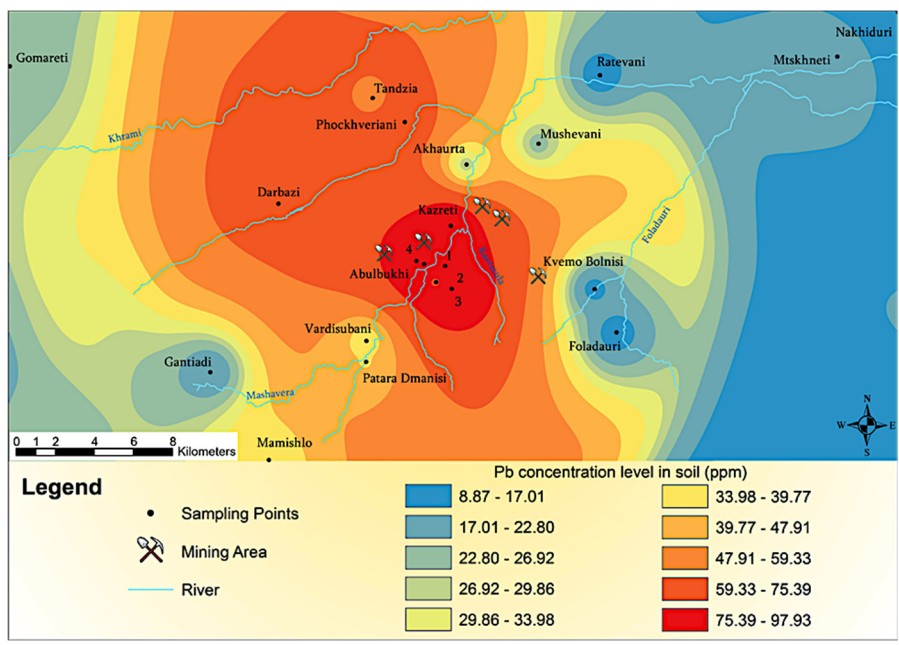

**Figure 9.** Distribution of lead in soils in the Bolnisi and Dmanisi municipalities. mg/kg. Numbers 1, 2, 3 and 4 are the sampling points in the village Balichi taken from the different locations (see the legend of the map on Figure 2), in the Table 3 they are indicated as Balichi 1, Balichi 2, Balichi 3, and Balichi 4 respectively.

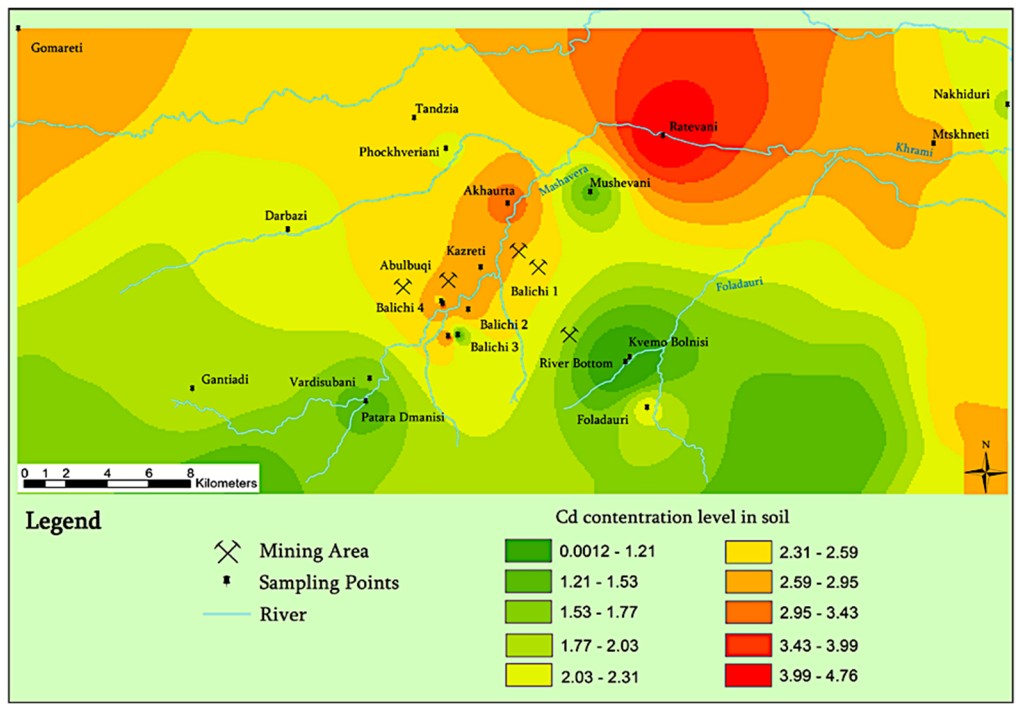

**Figure 10.** Distribution of cadmium in soils in the Bolnisi and Dmanisi municipalities, mg/kg. Numbers 1, 2, 3 and 4 are the sampling points in the village Balichi taken from the different locations (see the legend of the map on Figure 2), in the Table 3 they are indicated as Balichi 1, Balichi 2, Balichi 3, and Balichi 4 respectively.

Integrated water and soil pollution data in the study area revealed that in 2014, high concentrations of heavy metals in agricultural soil in the villages Nakhiduri and Mtskhneti resulted from using the Mashavera river water for irrigation, which was contaminated by various heavy metals from the wastewater flowing out the mining factory. Nakhiduri and Mtskhneti were at high Ecological Risk (ER) for Cd pollution in 2014. In 2017, the new study showed that trace metals concentration increased in the villages near the mining area, which is due to opening new mines and release of higher amounts of dust particles into the atmosphere during intensive open-pit mining, which caused soil pollution by Cd, Cu, Mn, Pb, and Zn. In 2017, in nearly every examined village, Cd pollution was at high Ecological Risk, the highest was observed in the village Ratevani. A high correlation was found between the concentration of Pb in the soils and the distance from these soils to the mining area, meaning that the closer the soil is to the mine, the higher the lead content is in it, and vice versa. No correlation was found for Cd, Cu, Mn, or Zn. Interpolation maps showed Pb and Cd distribution in the region, which supports the assumption that the highest Pb concentration is in the villages near the mining area, and the highest Cd content is in further places. Further investigations are recommended to be done to protect the human health of those living in this region.

## 4. Conclusions

Thus, mining activities cause water and soil pollution and impact environmental quality in the region. The mining area and ore production increase cause environmental quality decrease in the nearby villages. Using contaminated river water for soil irrigation causes soil pollution. Heavy metals are reduced in the fallow soils that were not harvested for four years, indicating the leaching of heavy metals to the deeper layers of soil. Seasonality impacts the fluctuations of heavy metals concentrations in the water. Extremely high ER and potential RI risk were observed in some of the investigated villages, which requires further research in the area to protect the life of the region's population.

**Author Contributions:** Conceptualization, A.G.; Methodology, G.A.; Resources, I.A. and L.A.; Software, M.A.; Supervision, L.M. and R.G.; Visualization, M.A.; Writing—original draft, M.A. All authors have read and agreed to the published version of the manuscript.

**Funding:** This research was funded by the Shota Rustaveli National Science Foundation of Georgia (grant numbers DO/187/150/14 and PHDF-21-1718) and Heinrich Böll Stiftung. Publication charges for this article were fully paid by the University Library System, University of Pittsburgh.

**Institutional Review Board Statement:** Not applicable.

**Informed Consent Statement:** Not applicable.

**Data Availability Statement:** All of the data used in out article is our own research. This data can be found here: http://www.jeb.co.in/journal_issues/202003_mar20_spl/paper_06.pdf, accessed on 21 March 2022.

**Acknowledgments:** Authors thank Daniel Joseph Bain and Ramos Caineta, Júlio for reviewing the article.

**Conflicts of Interest:** The authors declare no conflict of interest.

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
