# Peer review of "Mining-Related Metal Pollution and Ecological Risk Factors in South-Eastern Georgia"

_sustainability, doi:10.3390/su14095621_

Round 1
Reviewer 1 Report
Referee’s comments
General comment
In this study, heavy metal concentration in soil and river water around mining area were measured to evaluate potential ecological risks. The topic is important and suitable for the journal, however, the validity of the analytical results is not shown and the explanation of the results is not adequate. This paper should not be accepted in the current form.
Specific comments
Q1. Lines 128-130:
Each sample was filtered through filter paper and placed in the oven for the evaporation. Samples were digested for heavy metals analysis using Atomic Absorption Spectroscopy (AAS).
The digestion of water samples should be described in detail. What kind of filter was used?
Q2. 2.2 Elemental Analyses
There is no description about the reliability evaluation of the analysis value.
Q3. Table 1
Potential ecological risk (RI) → Potential ecological risk index (RI)
Ri < 150 → RI<150
Q4. Lines 133- 134
ER=Ti×Cf, where Ti is the Toxic-response factor for a given substance (see table 1),
Q5. Ti → Tri
table 1 → Table 1
The explanation of Toxic-response factor was not shown in Table 1.
Q5. Line 137
table 1 → Table 1
Q6. Table 1
Ecological risk factor, ER, Eir should be indicated by the same symbol.
Q7. Lines 141 – 147
No explanation is needed for the correlation coefficient.
Q8. Lines 177-178:
However, this study did not confirm this hypothesis to be acceptable for all the trace elements, that were analyzed, we will discuss this in the next section.
I do not understand this sentence.
Q9. Figs. 1, 2
It is better to have a scale in the figure to show the scale of the eruption.
Q10. Lines 198-205:
Since the precisely and accuracy of the analytical methods have not been shown, the differences between 2014 and 2017 may not be significant. The quantitative discussion has no meaning.
Q11. Table 4
Why were the As values shown in the Table 4?
Q12. Line 220
fig. 4 → Fig. 4
Q13. line 235
fig. 5 → Fig. 5
Q14. lines 260-264
This fact can be resulted by Tailings Pond, which is before the MA2 sampling point and is the source of acid mine drainage with pH values <3 and high dissolved heavy metals concentrations.
The authors should show the data about pH and dissolved heavy metals concentrations in the mine drainage.
Q15. Fig. 7
MA3 and MA4 were not shown in the figure.
Q16. Lines 303-304
In the case of cadmium, there was a negative correlation
-0.2<r<0.2 means no correlation.
Q17. Line 314
Mg/kg → mg/kg
over
Author Response
Dear Reviewer,
Thank you very much for reviewing our article.
We have edited the article according to your comments.
Sorry if we missed something, we will be happy to correct it.
Please see the attachment for our responses to your comments.
Best regards,
Authors

Reviewer 2 Report
The abstract is too long and needs to be shortened.
The introduction does not clearly address the study's innovations and issues.
Figure 1. It is necessary to insert a general reference map for readers likely to be unfamiliar with the study area. The water sampling locations should be marked on the map as well.
Please, replace the section "2.2 Elemental analyses" with "2.2. Sampling and analytical methods"
In soil analysis reports trace element concentrations are commonly measured on the fine earth fraction (i.e. soil particles less than 2 mm in size). The authors should justify the choice of the particle-size fraction they used for analysis (< 0.18 mm).
Methodology description is incomplete. What quality control procedures were applied to ensure accuracy of the analytical results? The results will have no significance if the errors are not indicated. What are the detection limits of the analytical techniques?
Calculation of the Contamination factor (Cf) shoud be based on local background concentrations rather than world average shale values of trace elements.
For the calculation of the Ecological Risk Factor (ERI), the authors have used the sedimentological toxic factors (the Sti value) and should use the toxic-response factor (Tri value) which accounts for both the toxic factor requeriment (the Sti value) and the sensitivity requeriment given by the bioproduction index (the BPI-value).
Results concerning air transport (section 3.1) are not based on experimental methods. Please consider removing this section or giving quantitative results of atmospheric bulk deposition.
Caption of Figure 5: Please, be more explicit.
Figure 8 is of poor quality and difficult to read.
Results and discussions can be combined into one section.
The Conclusions section is missing.I suggest to the authors to convey the main findings of the research.
Author Response

(The authors gave the same response as above.)

Reviewer 3 Report
The third author's initial is missing in his name.
L118 HNO3 (3 with subscript)
Elements are sometimes referred to by their names, others by their symbols ( symbols in alphabetical order would be easier).
Table 3. Permissible limits only according to "Order of the Minister of Labor, Health 196 and Social Affairs of Georgia, 2001" do not seem appropriate. Compare a little with other international values, background, etc.
I would have used Enrichment factors and other parameters, even risk assesment.
Author Response

(The authors gave the same response as above.)

Round 2
Reviewer 1 Report
This paper has been modified adequately following reviewer's comments.
Reviewer 2 Report
This article has already been revised and highly improved.
However, extensive editing of English language and style is still required.
Reviewer 3 Report
I think the article is now suitable for publication in its present form